# Biochemical Analysis to Understand the Flooding Tolerance of Mutant Soybean Irradiated with Gamma Rays

**DOI:** 10.3390/ijms25010517

**Published:** 2023-12-30

**Authors:** Setsuko Komatsu, Tiantian Zhou, Yuhi Kono

**Affiliations:** 1Faculty of Environment and Information Sciences, Fukui University of Technology, Fukui 910-8505, Japan; f20031ts@edu.fukui-ut.ac.jp; 2Central Region Agricultural Research Center, National Agriculture and Food Research Organization, Joetsu 943-0193, Japan; k41523@affrc.go.jp

**Keywords:** proteomics, mutant soybean, flooding tolerance, cell organization, protein degradation

## Abstract

Flooding stress, which reduces plant growth and seed yield, is a serious problem for soybean. To improve the productivity of flooded soybean, flooding-tolerant soybean was produced by gamma-ray irradiation. Three-day-old wild-type and mutant-line plants were flooded for 2 days. Protein, RNA, and genomic DNA were then analyzed based on oppositely changed proteins between the wild type and the mutant line under flooding stress. They were associated with cell organization, RNA metabolism, and protein degradation according to proteomic analysis. Immunoblot analysis confirmed that the accumulation of beta-tubulin/beta-actin increased in the wild type under flooding stress and recovered to the control level in the mutant line; however, alpha-tubulin increased in both the wild type and the mutant line under stress. Ubiquitin was accumulated and genomic DNA was degraded by flooding stress in the wild type; however, they were almost the same as control levels in the mutant line. On the other hand, the gene expression level of *RNase H* and *60S ribosomal protein* did not change in either the wild type or the mutant line under flooding stress. Furthermore, chlorophyll a/b decreased and increased in the wild type and the mutant line, respectively, under flooding stress. These results suggest that the regulation of cell organization and protein degradation might be an important factor in the acquisition of flooding tolerance in soybean.

## 1. Introduction

Soybean, which is one of the major agricultural crops, is sensitive to flooding stress [1]. In flooded soils, the plant growth and grain yield of soybean is significantly reduced [2]. Due to the rich protein and oil content in soybean seeds, soybean is susceptible to internal damage and uneven soil flattening, heavy rainfall, and poorly drained fields [3], resulting in impeded seed germination and seedling establishment. Plants frequently face hypoxic stress due to habitat changes, such as sudden floods, heavy rainfall, and low oxygen at high altitudes [4], though oxygen availability is essential for their survival. In soybean breeding or genetic studies, when soybean is exposed to flood stress at various growth stages, the vegetative period is reduced by 17–40% and the reproductive period is reduced by 40–57% due to flooding [5]. On the other hand, in Japan, soybean in the early growth stage is significantly suppressed due to flooding stress during the rainy season. The candidate genes could help researchers better understand the underlying physiological mechanisms of flooding stress tolerance in soybean in the early growth stage.

Floods cause excessive moisture to accumulate in crop roots and dramatically decrease oxygen levels in the soil [6]. Under flooding stress, soybean has displayed differential regulation of proteins involved in the suppression of reactive oxygen species scavenging, signal transduction, transcriptional regulation, sucrose accumulation/glucose degradation, alcohol fermentation, mitochondrial impairment, gamma-aminobutyric acid shunt, ubiquitin/proteasome-mediated proteolysis, and cell-wall loosening [7,8]. In particular, plants can cope with flooding conditions by adopting an orchestrated set of morphological adaptations and physiological adjustments controlled by an elaborate hormonal signaling network [9]. Ethylene transcription factors were found to be involved in regulating the response of plants to low-oxygen stress, and crop yields were improved under suboptimal growing conditions [10]. Ethylene transcription factors were highly induced by submergence and then downregulated during the recovery phase in soybean [11]. Although flooding response mechanisms in soybean were reported, the characterization of flooding tolerance mechanisms is needed regarding agricultural usage.

To characterize flooding tolerance mechanisms in soybean, flooding-tolerant mutant lines were generated by gamma-ray irradiation with six-fold tolerance screening [12]. Gel-based proteomic analysis using this mutant line indicated that the important factor for the acquisition of flooding tolerance in soybean is the activation of the fermentation system during the early stages of flooding [12]. Gel-free proteomics [13], RNA-seq transcriptomics [14], and metabolomics [15] were performed at the initial stage of flooding stress using the same mutant line and abscisic acid-treated soybean, which highlights flooding tolerance. These results indicated that the ethylene and abscisic acid signaling pathways, which may be involved in tolerance to initial flooding, prevented the growth-inhibitory effect of soybean under flooding [16]. On the other hand, how soybean achieves a harmonious relationship within these responses to initial flooding is elusive.

Furthermore, because the causative gene of this mutant line has not yet been identified, the previous mutant line [12], which showed flooding tolerance, was crossed with the parent cultivar Enrei. Using this mutant line, morphological and proteomic analyses were performed under flooding stress [17], indicating that the regulation of cell death through the fermentation system and glycoprotein folding was an important factor in the acquisition of flooding tolerance [17]. Although the importance of the fermentation system and glycoprotein folding against flooding tolerance was clarified using this mutant line, other cellular mechanisms were not identified. To obtain more comprehensive results regarding flooding tolerance, previous proteomic data from gene ontology analysis [17] were re-analyzed using MapMan bin codes. Based on the results, immunoblot, gene expression, and other analyses were further performed.

## 2. Results

### 2.1. Identification and Functional Investigation of Proteins in the Mutant Line under Flooding Stress

Three-day-old seedlings from the mutant line and the wild type were subjected to flooding for 2 days and proteins were analyzed using a gel-free/label-free proteomic technique [17]. Using gene ontology categorization, the abundance of oppositely changed proteins between these two kinds of soybean was associated with endoplasmic reticulum [17]. In this study, to obtain further information on the flooding tolerance mechanism, previous proteomic data [17] were re-analyzed using MapMan bin codes (Figure 1 and Figure 2, and Appendix A). Among the 127 proteins, 79 and 48 increased and decreased, respectively, in the mutant line compared with the wild type under non-flooding conditions (Figure 1A and Appendix A). Among the 85 proteins, 33 and 52 increased and decreased, respectively, in the mutant line compared with the wild type under flooding stress (Figure 1B and Appendix A). Using MapMan bin codes, newly analyzed categories in this study were associated with protein degradation, RNA metabolism, and transport in the mutant line compared with the wild type under flooding stress (Figure 1).

Among the 986 proteins, 514 and 472 increased and decreased, respectively, in the wild type under flooding stress compared with non-flooding conditions (Figure 2A and Appendix A). Among the 833 proteins, 350 and 483 increased and decreased, respectively, in the mutant line under flooding stress compared with non-flooding conditions (Figure 2B and Appendix A). Using MapMan bin codes, in this study, newly analyzed categories with opposite changes were associated with protein degradation, cell organization, RNA metabolism, minor CHO, co-factor, oxidative pentose phosphate, and the tricarboxylic acid cycle between the mutant line and the wild type under flooding stress (Figure 2). The proteins related to protein degradation, cell organization, and RNA metabolism were further confirmed using immunoblotting, RNA expression, and other analyses.

### 2.2. Analyses of Ubiquitin Accumulation and Genomic DNA Degradation in the Mutant Line under Flooding Stress

To better uncover the changes in proteins categorized as protein degradation, immunoblot analysis of ubiquitin was performed (Figure 3). In this study, among proteins related to protein degradation, RBR-type E3 ubiquitin transferase, ubiquitin conjugate 2 domain-containing protein, ubiquitin protease, ubiquitin, ubiquitin thioesterase, and proteasome increased and decreased in the wild type and mutant, respectively, under flooding stress. Proteins were extracted from the roots of the wild type and the mutant line treated with or without flooding stress. The Coomassie brilliant blue staining pattern was used as a loading control (Figure 3A and Appendix A). To investigate the changes in proteins categorized as protein degradation, the accumulation of ubiquitin was analyzed (Appendix A). Immunoblot analysis confirmed that ubiquitin accumulated in the wild type under flooding stress; however, this accumulation was recovered to the control level in the mutant line even if it was under flooding stress (Figure 3B).

To investigate the changes in genomic DNA degradation, its accumulation was analyzed (Figure 4). Genomic DNA was extracted from the roots of the wild type and the mutant line treated with or without flooding stress. DNA concentrations were analyzed for each sample (Figure 4A). Genomic DNA was degraded in the wild type under flooding stress; however, it was not degraded in the mutant line even under stress (Figure 4B). These results indicated that protein and DNA degradation was suppressed in the mutant line even when it was flooded.

### 2.3. Analysis of Cell-Organization-Related Proteins in the Mutant Line under Flooding Stress, Which Were Identified Using Immunoblot Analysis

To better uncover the change in cell-organization-related proteins in the mutant line under flooding stress, the accumulation of alpha-tubulin, beta-tubulin, and beta-actin was analyzed using immunoblot analysis (Figure 5). Proteins were extracted from the root and the hypocotyl of the mutant line and wild type treated with or without flooding. The Coomassie brilliant blue staining pattern was used as a loading control (Appendix A), and the accumulation of alpha-tubulin, beta-tubulin, and beta-actin was analyzed with antibodies (Appendix A). The accumulation of alpha-tubulin increased in both the wild type and the mutant line under flooding stress (Figure 5A). The accumulation of beta-tubulin and beta-actin increased in the wild type under flooding stress; however, they recovered to the control level in the mutant line even if it was under stress (Figure 5B,C). These results indicated that beta-tubulin and beta-actin were regulated in the mutant line for stress tolerance under flooding.

### 2.4. Analysis of Gene Expression Levels of RNA Metabolism-Associated Proteins in the Mutant Line under Flooding Stress

To better uncover the change in RNA metabolism-associated proteins in the mutant line under flooding stress, the gene expression levels of *RNase* and *60S ribosomal protein* were analyzed in both the wild type and the mutant line under flooding stress using polymerase chain reaction analysis. RNAs were extracted from the roots and the hypocotyl of the mutant line and wild type treated with or without flooding. Polymerase chain reaction analysis was performed for the gene expression of *RNase* and *60S ribosomal protein* (Figure 6A,B, Appendix A). The expression pattern of *18S rRNA* was used as an internal control (Figure 6C and Appendix A), and the expression of *RNase* and *60S ribosomal protein* was analyzed (Figure 6). The gene expression levels of *18S rRNA* and *RNase* did not change between the wild type and the mutant line with or without flooding stress. Although the gene expression level of *60S ribosomal protein* was downregulated by flooding stress, they did not change between the wild type and the mutant line under flooding stress (Figure 6).

### 2.5. Analysis of Chlorophyll Contents in the Mutant Line under Flooding Stress

To understand the role of the hypocotyl in the mutant line under flooding stress, the chlorophyll contents were analyzed as photosynthesis parameters (Figure 7). The contents of chlorophylls *a* and *b* significantly decreased under flooding stress; however, they recovered in the mutant line, even if it was under stress (Figure 7). These results indicated that photosynthesis was improved in the mutant line, even if it was under flooding conditions.

## 3. Discussion

### 3.1. Protein Degradation, Cell Organization, and RNA Metabolism Changes in the Mutant Line under Flooding Stress, Which Were Identified Using Proteomic Techniques

Compared to cultivated soybean (*Glycine max* L. Merr.), annual climbing soybean (*Glycine soja* Sieb. and Zucc.) exhibits high genetic diversity and was recognized as the ancestor of cultivated soybean [18,19]. Wild soybean was more waterlog-tolerant than cultivated soybean [20]; most especially, the wild soybean accession PI342618B has demonstrated high flooding tolerance [21]. Furthermore, the flooding tolerance of cultivated soybean was improved by introducing favorable alleles from wild soybean into elite cultivars [22]. Although wild soybean can be used as an important source of genetic variability, the creation of mutants by gamma-ray irradiation is also a useful method [12]. The growth of the mutant lines was better than that of the wild type even if it was flooded, although the growth of the wild type was significantly suppressed by this stress compared with non-flooding conditions [12,17]. Unfortunately, the causative gene of this mutant line has not yet been identified.

Using this flooding tolerance mutant line, proteomic analysis was performed. According to the gene ontology classification, oppositely changed proteins, which are abundant between the wild type and the mutant line, were associated with the endoplasmic reticulum under flooding stress [17]. Based on this result, the regulation of cell death through the fermentation system and glycoprotein folding was an important factor in the acquisition of flooding tolerance [17]. To obtain further information on the flooding-tolerant mechanism, previous proteomic data with gene ontology analysis [17] were re-analyzed using MapMan bin codes in this study. Using MapMan bin codes, newly analyzed categories, in this study, were associated with protein degradation, RNA metabolism, and transport in the mutant line compared with the wild type under flooding stress (Figure 1). Additionally, these categories with opposite changes were associated with protein degradation, cell organization, RNA metabolism, minor CHO, co-factor, oxidative pentose phosphate, and the tricarboxylic acid cycle between the mutant line and the wild type under flooding stress (Figure 2). These current studies suggest that the proteins related to protein degradation and RNA metabolism are important factors for the flooding tolerance of soybean. Although it was reported that the proteins related to cell organization such as tubulin were regulated in soybean under flooding stress [8], there are almost no reports to support this claim. Because proteins related to cell organization were identified in the mutant line in this study, these proteins might be associated with flooding tolerance in soybean.

### 3.2. Ubiquitin Is Accumulated and Genomic DNA Is Degraded by Flooding Stress in the Wild Type; However, They Recover to the Control Level in the Mutant Line

It has been previously reported that cell death increased and decreased in the wild type and the mutant line, respectively, under flooding stress with the Evans blue staining method [17]. In this study, because proteins related to the ubiquitin–proteasome system increased and decreased in the wild type and the mutant line, respectively, under flooding stress, the accumulation of ubiquitin was analyzed. The ubiquitin-proteasome system has been identified as key regulator that acts in concert to regulate core aspects of responses to hypoxia in plants [23]. Under flooding, plant-derived smoke, which shows flooding tolerance, induced the inhibition of the ubiquitin–proteasome pathway and led to sacrifice-for-survival-mechanism-driven degradation of the root tips in soybean [24]. The amount of ubiquitinated proteins in soybean roots decreased after flooding treatment and recovered to levels similar to controls after de-submergence [25]. N-terminal degradation signal-mediated processes with the ubiquitin–proteasome system in plants were implicated in the regulation of traits with potential agronomic importance, including the responses to flooding tolerance [26]. Protein degradation mediated by the ubiquitin–proteasome system might be central to this regulation, which enabled the accumulation of metabolites and promoted root development during recovery after flooding.

In the context of ubiquitin-like proteins, another important emerging player is autophagy. Autophagic programmed cell death is a complex and highly regulated degradative process, which acts as a survival pathway in response to cellular stress [27]. Autophagic programmed cell death of the meristematic cells has been implicated in the root-tip death of several species, such as pea and maize, when exposed to severe stress conditions [28]. Initiating a response to hypoxia requires genomic reprogramming. The latter depends on the activity of master transcriptional regulators, which coordinate gene expression in response to hypoxia [29]. In this study, to investigate the changes in genomic DNA degradation, its accumulation was analyzed. It was indicated that genomic DNA was reduced in the wild type under flooding stress; however, this reduction was recovered to the control level in the mutant line even if it was under stress (Figure 4). These results implied that protein and DNA degradation were recovered in the mutant line even if it was in flooding conditions. These results, along with previous reports, suggest that the mutant line might improve the quality of proteins and DNA; as a result, cell death is suppressed.

### 3.3. Cell-Organization-Related Proteins Were Regulated in the Mutant Line under Flooding Stress and Were Identified Using Immunoblot Analysis

In this study, beta-tubulin and beta-actin as cell-organization-related proteins were regulated in the mutant line for stress tolerance under flooding (Figure 5). The plant cytoskeleton, which includes microtubules and actin filaments, is a dynamic component of plant cells. It rapidly and dynamically recombines to adapt to the environment and maintain a growth state under external or internal stimuli [30]. The root system for the underground parts of plants is a direct organ for acquiring nutrients and water, as well as responding to environmental stimuli. The actin cytoskeleton is a point of integration not only in root growth and development but also in plant response to environmental stimuli [31]. Cytoskeleton-associated proteins are important regulatory molecules involved in the rearrangement of the actin cytoskeleton in response to environmental signals [32]. Among them, actin depolymerization factors are a major protein family for actin filament disassembly and play an important role in plant response to salt stress [33]. These results suggest that cell-organization-related proteins also have important roles in the flooding tolerance of soybean.

Using comparative RNA sequencing, genes were identified in biological processes, which were differentially regulated under drought and flooding stresses. These results indicate that these overlapping responses, which include the negative regulation of cellulose, tubulin, photosystem I/II, and chlorophyll biosynthesis, as well as the positive regulation of trehalose/sucrose metabolism, promote energy storage saving under both submergence and drought stresses [8]. In this study, the chlorophyll contents were analyzed as photosynthesis parameters (Figure 7). The contents of chlorophylls *a* and *b* significantly decreased under flooding stress; however, they recovered in the mutant line even if under stress (Figure 7). The expression of photosynthesis and chlorophyll synthesis-related genes were significantly reduced under drought and flooding stresses [34], which limit the metabolic processes and thus help prolong survival under extreme conditions. These results suggest that photosynthesis might be improved in the mutant line, even under flooding conditions, which might help it survive flooding stress.

## 4. Materials and Methods

### 4.1. Plant Material and Treatment

Flooding-tolerant founder mutant soybean was crossed with wild-type soybean (*Glycine max* L. cultivar Enrei). Mutant line 1386-6 (G2) with flooding tolerance selected from progeny was used [17]. Seeds were sterilized with a 2% sodium hypochlorite solution, washed twice with water, and sown in 400 mL of silica sand in a plastic case. Soybean plants were grown under white-fluorescent light (160 µmol m^−2^ s^−1^, 16 h light period/day) at 25 °C and 60% humidity for 3 days. To induce flooding stress, 3-day-old seedlings were soaked with additional water above the silica sand surface. After 2 days of treatment, seedlings were used for immunoblotting, RNA expression, and other analyses. Three independent experiments were performed as biological replicates of all experiments. This means that the seeds were sown on different days. In each plastic case, a total of 14 seeds were sown evenly.

### 4.2. Functional Categorization of Mass Spectrometry Data

Three-day-old seedlings of the mutant line and wild type were flooded for 2 days and proteins were analyzed using a gel-free/label-free proteomic technique [17]. For MS data, RAW data, peak lists, and result files were deposited into the ProteomeXchange Consortium [35] via the jPOST [36] partner repository with dataset identifier PXD024711. Proteins were categorized based on function using MapMan bin codes [37].

### 4.3. Protein Extraction and Immunoblot Analysis

The samples (500 mg) were ground with RIPA extraction buffer (Nacalai Tesque, Kyoto, Japan) at 4 °C using a mortar and pestle. SDS sample buffer (Bio-Rad, Hercules, CA, USA) was added to the protein extracts [38]. After the determination of protein concentration [39], quantified proteins (10 µg) were separated by electrophoresis on a 10% SDS-polyacrylamide gel. Coomassie brilliant blue staining was used as a loading control. For immunoblot analysis, proteins were transferred to a polyvinylidene difluoride membrane using a semidry-transfer blotter. This membrane was blocked in Bullet Blocking One reagent (Nacalai Tesque, Kyoto, Japan) for 5 min and cross-reacted with the primary antibodies for 30 min. As primary antibodies, anti-ubiquitin (Cosmo Bio, Tokyo, Japan), alpha-tubulin (Abcam, Cambridge, UK), beta-tubulin (Proteintech, Rosemont, IL, USA), and beta-actin (Proteintech) antibodies were used. After cross-reaction, the membrane was incubated with anti-rabbit IgG conjugated with horseradish peroxidase (Bio-Rad) as the secondary antibody for 30 min. Signals were detected using a TMB Membrane Peroxidase Substrate Kit (Seracare, Milford, MA, USA). The integrated density of the bands was calculated using Image J software (version 1.8; National Institutes of Health, Bethesda, MD, USA).

### 4.4. RNA Extraction and Polymerase Chain Reaction Analysis

The samples (500 mg) were snap-frozen in liquid nitrogen and ground into a powder using a mortar and pestle. Total RNA was isolated with an RNeasy Plant Mini Kit (Qiagen, Venlo, The Netherlands) according to the protocol from the manufacturer. First-strand cDNA was synthesized from total RNA (1 µg) with the iSuperscript Reverse Transcription Supermix (BioRad). Gene-specific primers for *18S rRNA*, *60S ribosomal protein*, and *RNaseH* were constructed with Primer3Plus software [40] and used to amplify the 200–500 bp regions (Appendix A). Polymerase chain reaction analysis was performed with Emerald Amp PCR Master Mix (Takara, Tokyo, Japan) as follows: 98 °C for 10 s, 60 °C for 30 s, and 72 °C for 30 s, for a total of 30 cycles. Amplified products with polymerase chain reaction were separated on a 3% agarose gel and stained with the Atlas ClearSight Gold DNA stain (BioAtlas, Tartu, Estonia). The integrated densities of the bands were calculated using Image J software.

### 4.5. Genomic DNA Extraction and Electrophoresis

A portion (0.5 g) of the samples was quickly frozen in liquid nitrogen and ground into a powder with a mortar and pestle. Genomic DNA was extracted using a Genomic DNA Extraction Kit (NucleoSpin Plant II: Macgrey-Nagel, Duren, Germany) following the protocol from the manufacturer [41]. The absorbance of DNA extraction was measured at 260 nm. The genomic DNA was separated by 2% agarose gel and stained with the Atlas ClearSight Gold DNA stain. The integrated densities of the bands were calculated using Image J software.

### 4.6. Contents of Chlorophylls a and b

A portion (0.5 g) of the samples was submerged in 1 mL of N,N-dimethylformamide for 16 h at 4 °C. The absorbance of chlorophylls *a* and *b* released in the solvent was measured at 663.8 nm and 646.8 nm. The contents of chlorophylls *a* and *b* were calculated using absorbance as follows: chlorophylls *a* and *b* (µM) = 19.4 × A_646.8_ + 8.05 × A_663.8_ [42,43].

### 4.7. Statistical Analysis

The statistical significance of data between the two groups was analyzed using Student’s *t*-test. A *p*-value of less than 0.05 was considered statistically significant.

## 5. Conclusions

To improve the productivity of flooded soybean, flooding-tolerant soybean was produced by gamma-ray irradiation. The main findings are as follows: (i) regarding cell organization, the accumulation of beta-tubulin/beta-actin increased in the wild type under flooding stress and recovered to the control level in the mutant line, whereas alpha-tubulin increased in both the wild type and the mutant line under stress; (ii) regarding protein degradation, ubiquitin was accumulated and genomic DNA was degraded by flooding stress in the wild type, but recovered to the control level in the mutant line; and (iii) regarding RNA metabolism, the gene expression levels of *RNase* and *60S ribosomal protein* did not change in either the wild type or the mutant line under flooding stress. These results suggest that the regulation of cell organization and protein degradation might be an important factor in the acquisition of flooding tolerance in soybean.

## Figures and Tables

**Figure 1 ijms-25-00517-f001:**
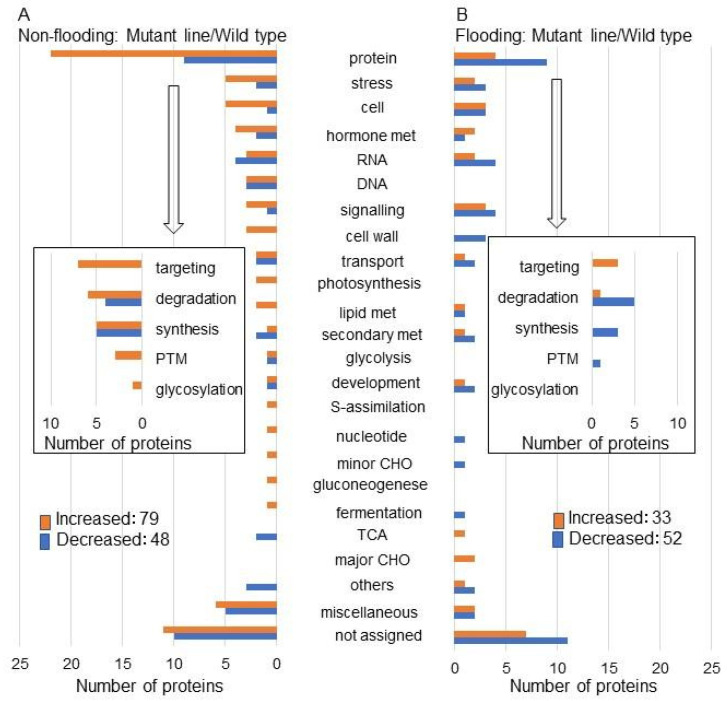
Functional categories of proteins with differential abundance in the mutant line compared with the wild type under flooding stress or non-flooding conditions. Functional categories of changed proteins were determined using MapMan bin codes (Appendix A). (**A**) Changed proteins in mutant-line compared with wild-type soybean under non-flooding conditions. (**B**) Changed proteins in mutant-line compared with wild-type soybean under flooding stress. Red and blue columns show increased and decreased proteins. Abbreviations: TCA, tricarboxylic acid cycle; PTM, posttranslational modification.

**Figure 2 ijms-25-00517-f002:**
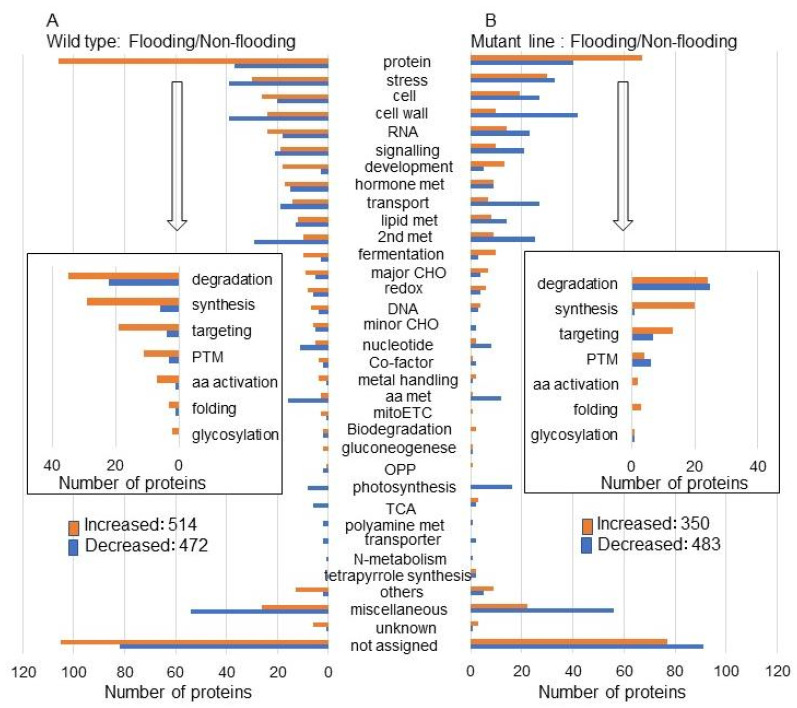
Functional categories of proteins with differential abundance in the mutant line or wild type under flooding stress compared with non-flooding conditions. Functional categories of changed proteins were determined using MapMan bin codes (Appendix A). (**A**) Changed proteins in wild-type soybean under flooding stress compared with non-flooding conditions. (**B**) Changed proteins in mutant-line soybean under flooding stress compared with non-flooding conditions. Red and blue columns show increased and decreased proteins. Abbreviations: TCA, tricarboxylic acid cycle; OPP, oxidative pentose phosphate; mitoETC, mitochondrial electron transport chain; PTM, posttranslational modification; and aa met, amino acid metabolism.

**Figure 3 ijms-25-00517-f003:**
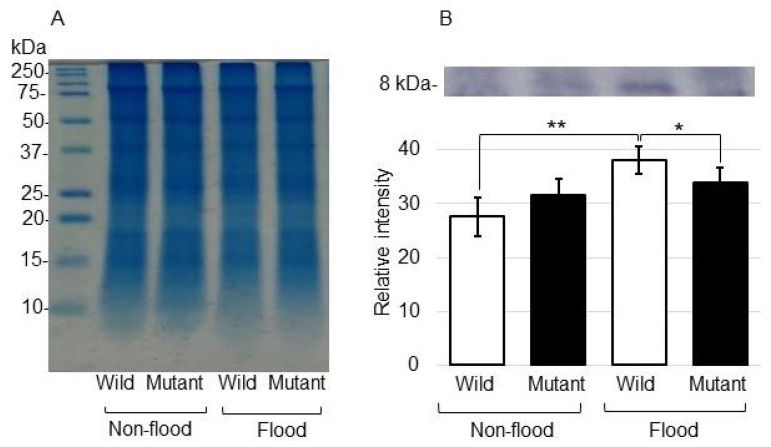
Immunoblot analysis of ubiquitin in the mutant line under flooding stress. Proteins were extracted from the roots, including hypocotyl, separated on SDS-polyacrylamide gel by electrophoresis and transferred onto a membrane. The membrane was cross-reacted with an anti-ubiquitin antibody. The Coomassie brilliant blue staining pattern was used as a loading control (**A**) (Appendix A). The integrated densities of the bands were calculated using ImageJ software (**B**). Data are shown as the means ± SD from 3 independent biological replicates (Appendix A). Student’s *t*-test was used to compare values between control and treatment as well as wild type and mutant line under flooding stress. Asterisks indicate a significant change (* *p* ≤ 0.05, ** *p* ≤ 0.01).

**Figure 4 ijms-25-00517-f004:**
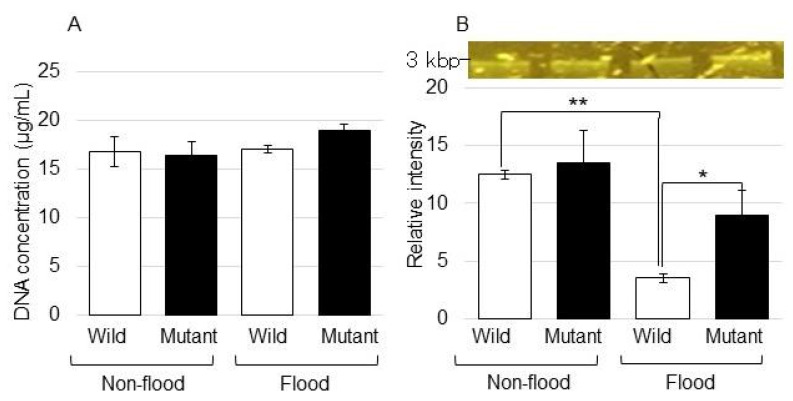
Analysis of genomic DNA degradation in the roots of the mutant line under flooding stress. After flooding stress, genomic DNA was extracted from the roots of the wild type and the mutant line. (**A**) The concentration of genomic DNA. (**B**) The pattern of agarose gel electrophoresis of extracted genomic DNA. Data are shown as the means ± SD from 3 independent biological replicates. Student’s *t*-test was used to compare values between control and treatment as well as wild type and mutant line under flooding stress. Asterisks indicate a significant change (* *p* ≤ 0.05, ** *p* ≤ 0.01).

**Figure 5 ijms-25-00517-f005:**
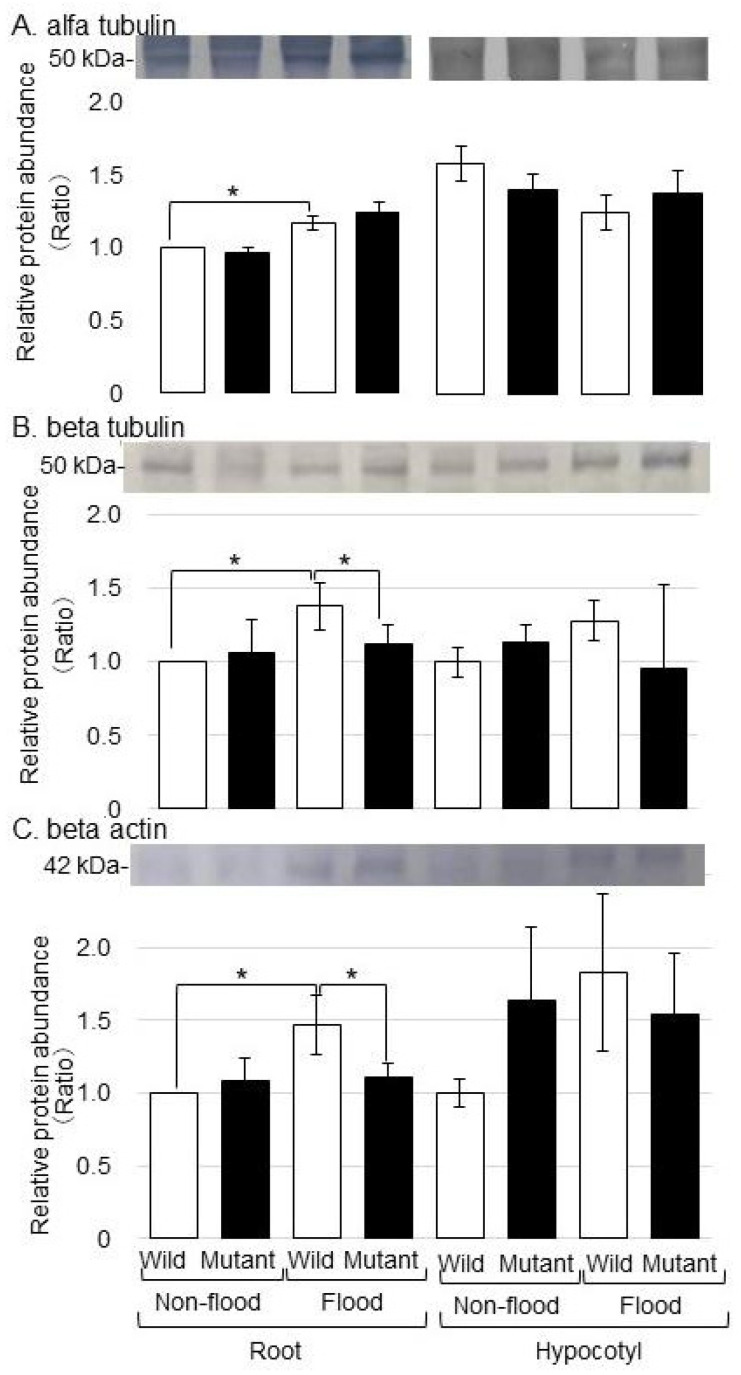
Immunoblot analysis of proteins involved in cell organization in the mutant line under flooding stress. Proteins were extracted from the root and hypocotyl, separated on SDS-polyacrylamide gel by electrophoresis and transferred onto a membrane. The membrane was cross-reacted with anti-alpha-tubulin, beta-tubulin, and beta-actin antibodies. The Coomassie brilliant blue staining pattern was used as a loading control (Appendix A). The integrated densities of the bands were calculated using ImageJ software with 3 independent biological replicates (Appendix A). Data are shown as the means ± SD from 3 independent biological replicates. Student’s *t*-test was used to compare values between control and treatment as well as wild type and mutant line under flooding stress. Asterisks indicate a significant change (* *p* ≤ 0.05).

**Figure 6 ijms-25-00517-f006:**
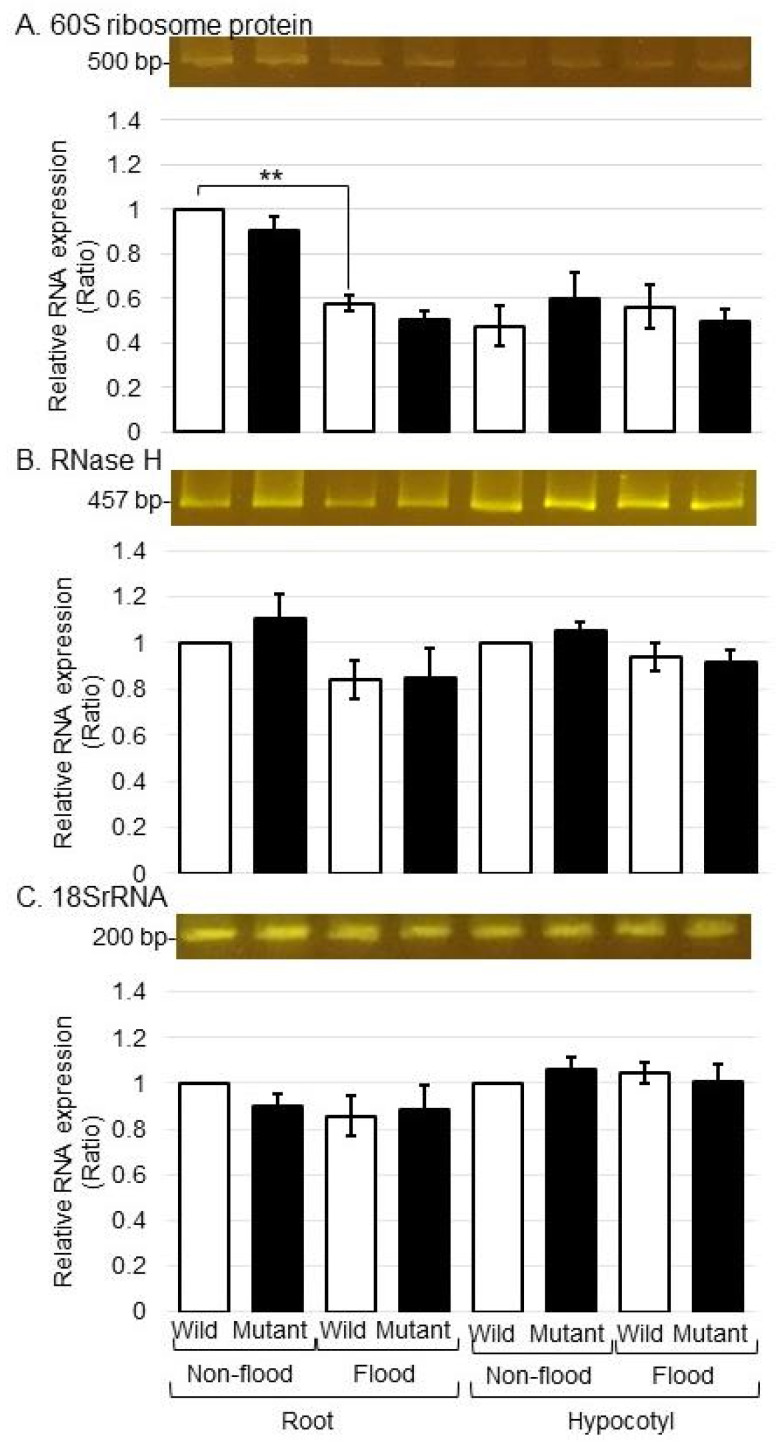
Gene expression of proteins involved in RNA metabolism in the mutant line under flooding stress. Gene expression analysis of *60S ribosome protein* and *RNase H* in the wild type and the mutant line with or without flooding stress was performed. (**A**) *60S ribosome protein*- and (**B**) *RNase H*-specific oligonucleotides were used to amplify transcripts from total RNA isolated from the roots and hypocotyl. *18S rRNA* was used as an internal control (**C**). Data are shown as the means ± SD from 3 independent biological replicates. Student’s *t*-test was used to compare values between control and treatment as well as wild type and mutant line under flooding stress (** *p* ≤ 0.01).

**Figure 7 ijms-25-00517-f007:**
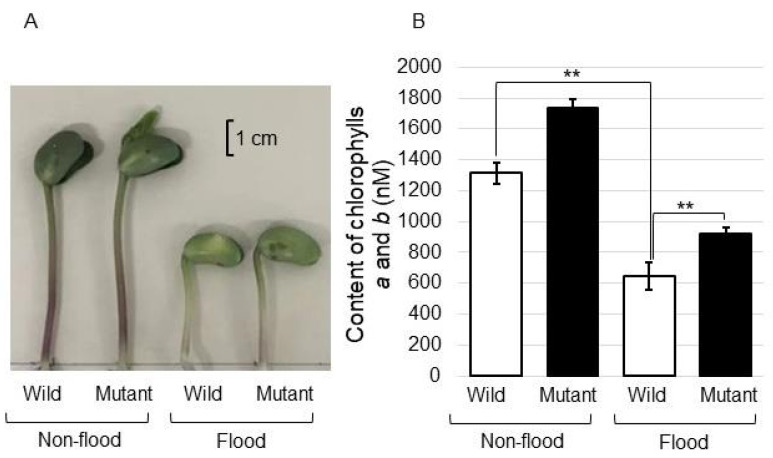
The contents of chlorophylls *a* and *b* in the hypocotyl of the mutant line under flooding stress. Photograph of the upper part of soybean plants with and without flooding stress (**A**). Size bar indicates 1 cm. Contents of chlorophylls a and b of the hypocotyl of soybean with and without flooding stress (**B**). Chlorophylls *a* and *b* extracted from the hypocotyl of the wild type and the mutant line were measured. Data are shown as the means ± SD from 3 independent biological replicates (Appendix A). Student’s *t*-test was used to compare values between control and treatment as well as wild type and mutant line under flooding stress. Asterisks indicate a significant change (** *p* ≤ 0.01).

## Data Availability

Data are contained within the article and Appendix A.

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
