# Peer review of "Biochemical Analysis to Understand the Flooding Tolerance of Mutant Soybean Irradiated with Gamma Rays"

_ijms, 2023, doi:10.3390/ijms25010517_

Round 1
Reviewer 1 Report
Comments and Suggestions for Authors
The authors made attempts to describe mechanism of flloding stress tolerance. However, authors presented only differetially expressed proteins and did not linked it with cell tzpe, physiology etc. The authors need to re-organise paper, change title and correct many statement.
Below are some details:
Title: not very suitable, two different things are mixed. There is no any direct link between cell organization (by the way, what does it mean, it is too general) and protein degradation.
Summary “what do you mean as three days old? “Three-day-old wild type” Seedlings? This object is too small to have any relation with real flooding -resistance mechanism and flooding effect at this stage have a little connection with real flooding on mature plants. This is effect of flooding on seeds germination/seedlings establishment.
Line 36: hypocotyl pigmentation??
Line 39: ”excess water in crop roots”??
Lines 47- 49: TF can not improved itself.
Line 60: “Based on these results, ethylene promoted soybean growth at initial flooding [16].” ?? This sentence is wrong. Please, reformulate.
Lines 63-67: is a very confusing part.
Line 75: “Three-day-old mutant line and wild type was subjected” = Three-day-old seedlings from the mutant line and wild type was subjected..
Lines 73- 74: there are not any information about cell organization. Instead, only information about differentially abundant proteins. Please, re-formulate correspondingly.
Lines 132- 138: please, re-write in convincing way.
“Genomic DNA was reduced in wild type under flooding stress; however, this reduction was recovered to the control level in mutant“ ? Genomic DANN can not reduced, it can degradated. Do you mean that in the mutant DANN recover for degradation? Plesae, write in correct way.
Figure 4: „ Evaluation of cell death“???? Where is cell death?? Cell death can be detected only in situ. Authors can use in situ nuclei label or TUNNELL assay to show degrataion. BUT not cell death.
Line 158: cell organisation is different in different cell type, Even sligtly significant changes in WB can not reflacted in realty cell organization. Please, confirmed cell organization changes in situ.
Line 206: „Chlorophyll Contents in Mutant Line Were Regulated under Flooding Stress“ ?? contemnts regulated in both wt and mutant lines. you meant different thing here.
Lines 321: „seedling case“??
Line 389: „mechanism“ ?? I have not seen any reliable machanism. Changes in 100 protein abundance is not a mechanism. The mechanism include chain of reaction and identification of trigger what induced these chains is the mechanism.
Moreover, mecahnism can be study only in situ since trigger activated first in a few key cells.
In addition, epigenetic changes under flooding stress serve as „integrator“ of seedlings response. This point is missing in the text.
Comments on the Quality of English LanguageModerate grammar.
Author Response
Reviewer 1
The authors made attempts to describe mechanism of flloding stress tolerance. However, authors presented only differetially expressed proteins and did not linked it with cell tzpe, physiology etc. The authors need to re-organise paper, change title and correct many statement. Below are some details:
Answer: Thank you very much for your critical comments. Based on your comments, this article has carefully been revised.
Title: not very suitable, two different things are mixed. There is no any direct link between cell organization (by the way, what does it mean, it is too general) and protein degradation.
Answer: Thank you very much for your suggestion. The title has been changed as follows: “Biochemical Analysis to Understand Flooding Tolerance of Mutant Soybean Irradiated with Gamma Rays”
Summary “what do you mean as three days old? “Three-day-old wild type” Seedlings? This object is too small to have any relation with real flooding -resistance mechanism and flooding effect at this stage have a little connection with real flooding on mature plants. This is effect of flooding on seeds germination/seedlings establishment.
Answer: Thank you very much for your comments and suggestion. As you pointed out, in soybean breeding or genetic studies, when soybeans are exposed to flood stress at various growth stages, the vegetative period is reduced by 17-40% and the reproductive period is reduced by 40-57% due to flooding (Nguyen et al., 2012). On the other hand, in Japan, soybean at early growth stage is significantly suppressed due to flooding stress during the rainy season. In this study, the response of proteins was evaluated to flooding at early growth stages. This explanation has been added to the introduction section in red.
(Nguyen, V.; Vuong, T.; VanToai, T.; Lee, J.; Wu, X.; Mian, M.; Nguyen, H. Mapping of quantitative trait loci associated with resistance to Phytophthora sojae and flooding tolerance in soybean. Crop. Sci. 2012, 52, 2481–2493.)
Line 36: hypocotyl pigmentation??
Answer: The sentence, which is including the word “hypocotyl pigmentation” has been deleted.
Line 39: ”excess water in crop roots”??
Answer: Thank you very much for your point out. This sentence has been corrected as follows: Floods cause excessive moisture to crop roots and dramatically decrease oxygen levels in the soil [6].
Lines 47- 49: TF can not improved itself.
Answer: We apologize for using an inappropriate expression. This sentence has been corrected as follows: “Ethylene transcription factors were involved in regulating the response of plants to low oxygen stress and crop yields were improved under suboptimal growing conditions”
Line 60: “Based on these results, ethylene promoted soybean growth at initial flooding [16].” ?? This sentence is wrong. Please, reformulate.
Answer: We are sorry we made a mistake, again. This sentence has been corrected as follows: “These results indicated that soybean growth was promoted through the ethylene signaling pathway at initial flooding.”
Lines 63-67: is a very confusing part.
Answer: We are sorry that this part was a confusing. This part has been corrected as follows: “Furthermore, because the causative gene of this mutant line has not yet been identified, the previous mutant line [12], which showed flooding tolerance, was crossed with parent cultivar Enrei. Using this mutant line, morphological and proteomic analyses were performed under flooding stress [17], indicating that the regulation of cell death through the fermentation system and glycoprotein folding was an important factor for the acquisition of flooding tolerance [17].”
Line 75: “Three-day-old mutant line and wild type was subjected” = Three-day-old seedlings from the mutant line and wild type was subjected..
Answer: Thank you very much for your correction. As suggested, this sentence has been corrected as follows: “Three-day-old seedlings from the mutant line and wild type were subjected”
Lines 73- 74: there are not any information about cell organization. Instead, only information about differentially abundant proteins. Please, re-formulate correspondingly.
Answer: We are sorry this inconvenient matter. The title of section “2.1” has been changed as follows: “Identification and Functional Investigation of Proteins in Mutant Line under Flooding Stress.”
Lines 132- 138: please, re-write in convincing way.
“Genomic DNA was reduced in wild type under flooding stress; however, this reduction was recovered to the control level in mutant“ ? Genomic DANN can not reduced, it can degradated. Do you mean that in the mutant DANN recover for degradation? Plesae, write in correct way.
Answer: Again we are sorry that we made a mistake. This paragraph has been corrected as follows: “Genomic DNA was degradated in wild type under flooding stress; however, it was not degradated in mutant line even under stress (Figure 4B). These results indicated that protein and DNA degradation was suppressed in the mutant line even when it was flooded.”
Figure 4: „ Evaluation of cell death“???? Where is cell death?? Cell death can be detected only in situ. Authors can use in situ nuclei label or TUNNELL assay to show degrataion. BUT not cell death.
Answer: Thank you very much for your comments and suggestion. The titles of the section “2.2” and the legend of Figure 4 have been corrected in red.
Line 158: cell organisation is different in different cell type, Even sligtly significant changes in WB can not reflacted in realty cell organization. Please, confirmed cell organization changes in situ.
Answer: Thank you very much for your critical comments. Based on your suggestion, we should analyze cellular tissue using in situ hybridization in the next step. Unfortunately, we currently do not have this system in our lab.
Line 206: „Chlorophyll Contents in Mutant Line Were Regulated under Flooding Stress“ ?? contemnts regulated in both wt and mutant lines. you meant different thing here.
Answer: as suggested, because this meaning was different thing, the title of “2.5” has been corrected in red.
Lines 321: „seedling case“??
Answer: The word “seedling case” has been changed to “plastic case”.
Line 389: „mechanism“ ?? I have not seen any reliable machanism. Changes in 100 protein abundance is not a mechanism. The mechanism include chain of reaction and identification of trigger what induced these chains is the mechanism. Moreover, mecahnism can be study only in situ since trigger activated first in a few key cells. In addition, epigenetic changes under flooding stress serve as „integrator“ of seedlings response. This point is missing in the text.
Answer: Thank you very much for your guidance. Because we agree with your comments, all the words “mechanism” has been removed from this text. After analysis with cellular tissue using in situ hybridization in the next step, we will use the word of “mechanism”.
Reviewer 2 Report
Comments and Suggestions for Authors
The manuscript title “The Regulation of Cell Organization and Protein Degradation Relates to the Acquisition of Flooding Tolerance in Soybean” has scientific worth and this research enhances our understanding/ scientific knowledge about the mechanism of flood tolerance in Soybean. The presentation of this MS is good. I have some minor suggestions for authors:
Reviewer Comments:
1- In figure 6: why the level of significance was not added in figure 6B and C. Is the results are non-significant?? The bards are representing standard error or standard deviations in the graphs?? It is not mentioned in figure legends.
2- Please add some background of this study in the last paragraph of introduction section and add some scientific gap that needs to be filled.
Comments on the Quality of English LanguageThe manuscript title “The Regulation of Cell Organization and Protein Degradation Relates to the Acquisition of Flooding Tolerance in Soybean” has scientific worth and this research enhances our understanding/ scientific knowledge about the mechanism of flood tolerance in Soybean. The presentation of this MS is good. I have some minor suggestions for authors:
Reviewer Comments:
1- In figure 6: why the level of significance was not added in figure 6B and C. Is the results are non-significant?? The bards are representing standard error or standard deviations in the graphs?? It is not mentioned in figure legends.
2- Please add some background of this study in the last paragraph of introduction section and add some scientific gap that needs to be filled.
Author Response
Reviewer 2
The manuscript title “The Regulation of Cell Organization and Protein Degradation Relates to the Acquisition of Flooding Tolerance in Soybean” has scientific worth and this research enhances our understanding/ scientific knowledge about the mechanism of flood tolerance in Soybean. The presentation of this MS is good. I have some minor suggestions for authors:
Answer: Thank you very much for your suggestion. Based on your suggestion, this article has been improved.
1- In figure 6: why the level of significance was not added in figure 6B and C. Is the results are non-significant?? The bards are representing standard error or standard deviations in the graphs?? It is not mentioned in figure legends.
Answer: Thank you very much for your comments. As you point out, they were not significantly different in figure 6B and C. As suggested, these explanation has been added in the result section “2.4” in red as follows. Gene-expression level of 18S rRNA and RNase did not change between wild type and mutant line as well as between with and without flooding stress. Although gene-expression level of 60S ribosomal protein was downregulated by flooding stress, they did not change between both wild type and mutant line under flooding stress (Figure 6).
Furthermore, the explanation of standard deviations has been marked in the legends of Figure 6 in red as follows: Data are shown as the means ± SD from 3 independent biological replicates. Student’s t-test was used to compare values between control and treatment as well as wild type and mutant line under flooding stress (** p < 0.01).
2- Please add some background of this study in the last paragraph of introduction section and add some scientific gap that needs to be filled.
Answer: Thank you very much for your suggestion. The background of this study and the scientific gap that needs to be filled have been added in red as follows: Although the importance of the fermentation system and glycoprotein folding against flooding tolerance was clarified using this mutant line, it was not identified other cellular mechanisms. To obtain more comprehensive mechanisms regarding flooding-tolerance mechanisms, previous proteomic data from gene-ontology analysis [17] were re-analyzed using MapMan bin codes.
Round 2
Reviewer 1 Report
Comments and Suggestions for Authors
Thank you for corrections. There are some more still required.
Line 40: "molecular mechanisms of flooding stress tolerance" - here is common mistake. Molecular mechanism of the multicelluar organism can nkt be described because it is different for different cell type. You can only describe physiological mechnanism.
Line 57: six-fold, not time
line 63: "was promoted through the ethylene and abscisic-acid signaling pathways at initial flooding" ? It is not true. pathways possible involved in resisitance. In addition, it will be more precise to write prevent inhibitiry effect on the growth under flooding.
Line 331: "To induce flooding stress, 3-day-old seedlings were soaked 331
with additional water above the silica sand surface for immuno-blot, RNA expression, and other analyses."? It will be great to use two sentences here.
Comments on the Quality of English Language
Some sentences required minor editing.
Author Response
Reviewer 1
Thank you for corrections. There are some more still required.
Answer: Thank you very much for your additional kind correction.
Line 40: "molecular mechanisms of flooding stress tolerance" - here is common mistake. Molecular mechanism of the multicelluar organism can nkt be described because it is different for different cell type. You can only describe physiological mechnanism.
Answer: We could understand your comments. This word has been corrected in blue as follows: “physiological mechanisms of flooding stress tolerance”
Line 57: six-fold, not time
Answer: Thank you very much for your correction. As suggested, this ford has been corrected in blue.
line 63: "was promoted through the ethylene and abscisic-acid signaling pathways at initial flooding" ? It is not true. pathways possible involved in resisitance. In addition, it will be more precise to write prevent inhibitiry effect on the growth under flooding.
Answer: Thank you very much for your correction. This sentence has been corrected in blue as follows: “These results indicated that the ethylene and abscisic-acid signaling pathways, which may be involved in tolerance at initial flooding, prevented the growth-inhibitory effect of soybean under flooding.”
Line 331: "To induce flooding stress, 3-day-old seedlings were soaked 331
with additional water above the silica sand surface for immuno-blot, RNA expression, and other analyses."? It will be great to use two sentences here.
Answer: As suggested, this sentence was corrected as follows: “To induce flooding stress, 3-day-old seedlings were soaked with additional water above the silica sand surface. After 2 days of treatment, seedlings were used for immuno-blot, RNA expression, and other analyses.”